# Outcomes of High-Grade Cervical Dysplasia with Positive Margins and HPV Persistence after Cervical Conization

**DOI:** 10.3390/vaccines11030698

**Published:** 2023-03-18

**Authors:** Andrea Giannini, Violante Di Donato, Francesco Sopracordevole, Andrea Ciavattini, Alessandro Ghelardi, Enrico Vizza, Ottavia D’Oria, Tommaso Simoncini, Francesco Plotti, Jvan Casarin, Tullio Golia D’Augè, Ilaria Cuccu, Maurizio Serati, Ciro Pinelli, Alice Bergamini, Barbara Gardella, Andrea Dell’Acqua, Ermelinda Monti, Paolo Vercellini, Giovanni D’Ippolito, Lorenzo Aguzzoli, Vincenzo Dario Mandato, Luca Giannella, Cono Scaffa, Antonino Ditto, Francesca Falcone, Chiara Borghi, Mario Malzoni, Alessandra Di Giovanni, Maria Giovanna Salerno, Viola Liberale, Biagio Contino, Cristina Donfrancesco, Michele Desiato, Anna Myriam Perrone, Pierandrea De Iaco, Simone Ferrero, Giuseppe Sarpietro, Maria G. Matarazzo, Antonio Cianci, Stefano Cianci, Sara Bosio, Simona Ruisi, Lavinia Mosca, Raffaele Tinelli, Rosa De Vincenzo, Gian Franco Zannoni, Gabriella Ferrandina, Marco Petrillo, Giampiero Capobianco, Annunziata Carlea, Fulvio Zullo, Barbara Muschiato, Stefano Palomba, Stefano Greggi, Arsenio Spinillo, Fabio Ghezzi, Nicola Colacurci, Roberto Angioli, Pierluigi Benedetti Panici, Ludovico Muzii, Giovanni Scambia, Francesco Raspagliesi, Giorgio Bogani

**Affiliations:** 1Department of Gynecological, Obstetrical and Urological Sciences, “Sapienza” University of Rome, 00161 Rome, Italy; andre.giannini@uniroma1.it (A.G.); violante.didonato@uniroma1.it (V.D.D.); tullio.goliadauge@uniroma1.it (T.G.D.); ilaria.cuccu@uniroma1.it (I.C.); pierluigi.benedettipanici@uniroma1.it (P.B.P.); ludovico.muzii@uniroma1.it (L.M.); 2Gynecological Oncology Unit, Centro di Riferimento Oncologico-National Cancer Institute, 33081 Aviano, Italy; fsopracordevole@cro.it; 3Woman’s Health Sciences Department, Gynecologic Section, Polytechnic University of Marche, 60126 Ancona, Italy; a.ciavattini@univpm.it (A.C.); luca.giannella@ospedaliriuniti.marche.it (L.G.); 4Azienda Usl Toscana Nord-Ovest, UOC Ostetricia e Ginecologia, Ospedale Apuane, 54100 Massa, Italy; alessandro.ghelardi@uslnordovest.toscana.it; 5Gynecologic Oncology Unit, Department of Experimental Clinical Oncology, IRCCS “Regina Elena” National Cancer Institute, 00144 Rome, Italy; enrico.vizza@ifo.gov.it; 6Department of Woman’s and Child’s Health, Obstetrics and Gynecological Unit, San Camillo-Forlanini Hospital, 00152 Rome, Italy; ottavia.doria@uniroma1.it (O.D.); salerno.giovannamaria@gmail.com (M.G.S.); 7Department of Clinical and Experimental Medicine, University of Pisa, 56126 Pisa, Italy; tommaso.simoncini@med.unipi.it; 8Department of Obstetrics and Gynecology, Campus Bio-Medico University of Rome, 00128 Rome, Italy; f.plotti@unicampus.it (F.P.); r.angioli@unicampus.it (R.A.); 9Department of Obstetrics and Gynecology, ‘Filippo Del Ponte’ Hospital, University of Insubria, 21100 Varese, Italy; jvancasarin@gmail.com (J.C.);; 10Ospedale di Circolo Fondazione Macchi, 21100 Varese, Italyfabio.ghezzi@uninsubria.it (F.G.); 11Department of Obstetrics and Gynecology, IRCCS Ospedale San Raffaele, 20100 Milano, Italy; bergamini.alice@hsr.it; 12IRCCS S. Matteo Foundation, Department of Clinical, Surgical, Diagnostic and Paediatric Sciences, University of Pavia, 27100 Pavia, Italy; barbara.gardella@unipv.it (B.G.); arsenio.spinillo@unipv.it (A.S.); 13Gynaecology Unit, Fondazione IRCCS Ca’ Granda Ospedale Maggiore Policlinico, 20122 Milan, Italy; andrea.dellacqua1@gmail.com (A.D.); ermelinda.monti@policlinico.mi.it (E.M.); paolo.vercellini@policlinico.mi.it (P.V.); 14Division of Obstetrics and Gynecology, Cesare Magati Hospital, Azienda Unità Sanitaria Locale-IRCCS di Reggio Emilia, 42019 Scandiano, Italy; giovanni_dippolito@yahoo.it (G.D.); lorenzo.aguzzoli@ausl.re.it (L.A.); dariomandato@gmail.com (V.D.M.); 15Gynecology Oncology Unit, Istituto Nazionale Tumori IRCCS “Fondazione G. Pascale”, 80131 Naples, Italy; c.scaffa@istitutotumori.na.it (C.S.); francesca.falcone@istitutotumori.na.it (F.F.); s.greggi@istitutotumori.na.it (S.G.); 16Gynecological Oncology Unit, Fondazione IRCCS Istituto Nazionale dei Tumori di Milano, 20133 Milan, Italy; antonino.ditto@istitutotumori.mi.it (A.D.); raspagliesi@istitutotumori.mi.it (F.R.); 17Department of Obstetrics and Gynecology, S. Anna University Hospital, 44121 Ferrara, Italy; chiara.borghi@unife.it; 18Endoscopica Malzoni, Center for Advanced Endoscopic Gynecological Surgery, 83100 Avellino, Italy; malzonimario@gmail.com (M.M.); dott.a.digiovanni@gmail.com (A.D.G.); 19Department of Obstetrics and Gynecology, Ospedale Maria Vittoria, 10144 Torino, Italy; viola.liberale@gmail.com (V.L.); b.contino@iol.it (B.C.); 20Department of Obstetrics and Gynecology, Azienda ASL Frosinone, Ospedale S Trinità di Sora, 03039 Sora, Italy; cristina.donfrancesco@gmail.com (C.D.); micheledesiato@libero.it (M.D.); 21Gynecologic Oncology Unit, Sant’Orsola-Malpighi Hospital, 40138 Bologna, Italy; perronemyriam@gmail.com (A.M.P.); pierandrea.deiaco@unibo.it (P.D.I.); 22Academic Unit of Obstetrics and Gynaecology, IRCCS Ospedale Policlinico San Martino, 16132 Genova, Italy; simoneferrero@me.com; 23Department of Neurosciences, Rehabilitation, Ophthalmology, Genetics, Maternal and Child Health (DiNOGMI), University of Genova, 16132 Genova, Italy; 24Department of General Surgery and Medical Surgical Specialties, Gynecological Clinic University of Catania, Via S. Sofia 78, 95124 Catania, Italy; sarpietrogiuseppe@gmail.com (G.S.); mariagraziamatarazzo@gmail.com (M.G.M.); acianci@unict.it (A.C.); 25Department of Gynecologic Oncology, Università degli Studi di Messina, Policlinico G. Martino, 98122 Messina, Italy; stefanoc85@hotmail.it; 26San Paolo Hospital, Università degli Studi di Milano, 20142 Milan, Italy; sarabos2004@hotmail.it (S.B.); simona.ruisi13@gmail.com (S.R.); 27Department of Woman, Child and General and Specialized Surgery, University of Campania “Luigi Vanvitelli”, 80138 Naples, Italy; laviniamosca@gmail.com (L.M.); nicola.colacurci@unicampania.it (N.C.); 28Department of Obstetrics and Gynecology, “Valle d’Itria” Hospital, Martina Franca, Via San Francesco da Paola, 74015 Taranto, Italy; raffaeletinelli@gmail.com; 29UOC Ginecologia Oncologica, Dipartimento per la Salute Della Donna e del Bambino e Della Salute Pubblica, Fondazione Policlinico Universitario A. Gemelli, IRCCS, 00168 Rome, Italy; rosa.devincenzo@unicatt.it (R.D.V.); gianfranco.zannoni@policlinicogemelli.it (G.F.Z.); gabriella.ferrandina@gmail.com (G.F.); giovanni.scambia@policlinicogemelli.it (G.S.); 30Gynecologic and Obstetric Unit, Department of Medical, Surgical and Experimental Sciences, University of Sassari, 07100 Sassari, Italy; marco.petrillo@gmail.com (M.P.); capobia@uniss.it (G.C.); 31Department of Neuroscience, Reproductive Science and Dentistry, School of Medicine, University of Naples Federico II, 80131 Naples, Italy; nunziacarlea@gmail.com (A.C.); fulvio.zullo@unina.it (F.Z.); 32Studio Medico Muschiato, Loano & Finale Ligure, 17025 Loano, Italy; info@studiomedicomuschiato.it; 33Unit of Obstetrics and Gynecology, GOM of Reggio Calabria & University ‘Magna Graecia’ of Catanzaro, 88100 Catanzaro, Italy; stefanopalomba@tin.it

**Keywords:** HPV, conization, positive margins, HPV persistence

## Abstract

The objective of this work is to assess the 5-year outcomes of patients undergoing conization for high-grade cervical lesions that simultaneously present as risk factors in the persistence of HPV infection and the positivity of surgical resection margins. This is a retrospective study evaluating patients undergoing conization for high-grade cervical lesions. All patients included had both positive surgical margins and experienced HPV persistence at 6 months. Associations were evaluated with Cox proportional hazard regression and summarized using hazard ratio (HR). The charts of 2966 patients undergoing conization were reviewed. Among the whole population, 163 (5.5%) patients met the inclusion criteria, being at high risk due to the presence of positive surgical margins and experiencing HPV persistence. Of 163 patients included, 17 (10.4%) patients developed a CIN2+ recurrence during the 5-year follow-up. Via univariate analyses, diagnosis of CIN3 instead of CIN2 (HR: 4.88 (95%CI: 1.10, 12.41); *p* = 0.035) and positive endocervical instead of ectocervical margins (HR: 6.44 (95%CI: 2.80, 9.65); *p* < 0.001) were associated with increased risk of persistence/recurrence. Via multivariate analyses, only positive endocervical instead of ectocervical margins (HR: 4.56 (95%CI: 1.23, 7.95); *p* = 0.021) were associated with worse outcomes. In this high-risk group, positive endocervical margins is the main risk factor predicting 5-year recurrence.

## 1. Introduction

Human papillomavirus (HPV) is one of the most common sexually transmitted diseases worldwide [1]. HPV is not a single entity but a group of several types of viruses. More than 200 HPV types are known, and over 50 different types are known to cause genital infections in both men and women.

The different HPV types are classified into “oncogenic” and “non-oncogenic”, according to the capacity to integrate their own genetic material into the DNA of the host cells [2]. The International Agency of Cancer Research (IARC) identified 13 high-risk (HR) HPV types, classified as oncogenic. HR-HPV types include the alpha-5 type 51; alpha-6 types 56 and 66; alpha-7 types 18, 39, 45 and 59; and alpha-9 types 16, 31, 33, 35, 52 and 58 [3,4].

Risk factors related to the transmission of the virus are mostly related to sexual habits: unprotected sexual practices, multiple partners and the onset of sexual activity at an early age. However, failure to vaccinate before the onset of sexual activity, immunosuppression and exposure to other sexually transmitted diseases also increase the possibility of contracting the infection and developing cervical lesions. In the male, the development of precancerous lesions is an ascending problem, particularly among populations exposed to a higher risk of contracting the infection (i.e., young age, homosexuals).

Most HPV infections regress spontaneously without sequalae [5], but in a small percentage of patients, persistent HR-HPV infections could lead to the development of precancerous (intraepithelial neoplasia) and/or invasive cancerous lesions not only in the cervix but also in other body districts (including the whole lower genital tract, the anus and the head-neck district) [6,7,8,9]. The uterine cervix remains the most anatomically frequent site involved in HPV-related pre-invasive or invasive lesions [10,11]. There is strong evidence supporting that high-grade cervical lesions (CIN2-3) might evolve to cancer. Surgical treatment (i.e., excision) is the mainstay of treatment for CIN2-3 [12,13].

The risk of recurrence after surgical treatment for CIN 2-3 is not negligible, ranging between 5 and 10% at 5 years [6,14]. Several investigations attempted to assess the risk factors for cervical dysplasia persistence/recurrence, highlighting that positive surgical cervical margins and HR-HPV persistence are the main factors predicting the risk of recurrence [12,15,16,17,18,19,20]. The available data evaluated those two risk factors separately. Although patients with both positive surgical margins and HPV persistence are considered at high risk, no study investigated the outcomes of patients with both risk factors. In the present study, we aim to bridge this gap, thus assessing the outcomes of patients who have positive surgical margins and are experiencing HPV persistence.

## 2. Materials and Methods

We performed a multicenter retrospective study. Data of consecutive patients with newly diagnosed high-grade cervical dysplasia (HSIL/CIN2/CIN3), treated in Italy between 1 January 2010 and 31 December 2014, were collected into a dedicated database. The Institutional Review Board was obtained.

Inclusion criteria were: (i) histologically confirmed diagnosis of high-grade cervical dysplasia (CIN 2-3); (ii) the execution of cervical conization; (iii) description of margins status, both endocervical and ectocervical, at histology; (iv) persistence of the HPV infection detected during a pap smear performed 6 months after the surgical treatment; and (v) 5-year follow-up (for patients without recurrence, instead, patients with recurrence were included even if they did not complete the 5-year follow-up).

Patients excluded from the study were: (i) patients under the age of 18 years, (ii) those who had not provided informed consent for participation, (iii) those undergoing therapeutic procedures other than those analyzed in the present study, (iv) those suffering from invasive carcinoma at the time of conization, (v) those suffering from glandular lesions, (vi) those with an ongoing pregnancy and (vii) those who had had a previous total hysterectomy.

The primary endpoint of the study was to assess the outcomes of patients considered at high risk of recurrence (those with positive margins and persistent HPV infection). As secondary outcomes, we tried to identify predictive factors that might influence the risk of developing persistence and recurrence in this high-risk patients’ group.

Data collected included patients’ demographics, baseline characteristics, viral genotype, surgical treatment mode and status of surgical resection margins, persistence of HPV infection after conization (at 6 and 12 months), vaccination status as well as follow-up after surgery. Although patients were treated in different centers and by different surgeons, there were no differences in terms of patient care and services’ facilities over the study period.

Data were collected into dedicated databases updated on a regular basis by residents and trained nurses. All patients included underwent conization with direct colposcopic guidance. All patients underwent an outpatient follow-up at 6 and 12 months after surgery. Follow-up included: medical examination, outpatient colposcopic evaluation and p ap smear. Methods used for the detection of HR-HPV before and after treatment differed between centers (they included hybrid capture (HC2^®^), Cobas^®^, Clart^®^ [21]). No significant difference in the positivity rate was observed between centers. For the purpose of this study, we aim to evaluate the risk of developing persistent and recurrent high-grade cervical dysplasia. Persistence was defined as the detection of high-grade cervical lesions at the time of first follow-up visit. Recurrence was defined by the presence of at least one negative clinical evaluation between surgical treatment and the diagnosis of new CIN2+. For the purpose of the present study, low grade lesions (e.g., LSIL, ASCUS, CIN 1) were not considered as recurrent diseases. Duration of follow-up was counted from the date of the first conization and the date of last follow-up or secondary conization or hysterectomy. Written informed consent regarding tissue and data use for scientific purposes was obtained from all participating patients. Data transfer and use for statistical analyses were done in an anonymized fashion.

Basic descriptive statistics were used to describe the study population. The risk of developing recurrence was evaluated using Kaplan-Meir and Cox proportional hazard regression models. Hazard ratio (HR) and 95% confidence intervals (CI) were calculated for each comparison. Univariate and multivariate analyses were performed when appropriate. Statistical analyses were performed using GraphPad Prism version 6.0 (GraphPad Software, San Diego, CA, USA), IBM-Microsoft SPSS (SPSS Statistics. International Business Machines Corporation IBM 2013 Armonk, NY, USA) version 20.0

## 3. Results

Overall, 2966 patients underwent conization for CIN2-3 during the study period. All patients were screened to identify HPV genotype(s) before conization. Figure 1 shows the flow of patients thorough the study design. Among the whole population, 163 (5.5%) patients met the inclusion criteria, being at high risk due to the presence of positive surgical margins and experiencing HPV persistence. The median age of the study population was 42.8 years (range, 20–74) and the median of body mass index was 24.8 kg/m^2^ (range, 18.4–42.0). Among the 163 patients, 102 (63%) and 73 (44.7%) had a diagnosis of CIN 3 and were HPV 16/18 positive, respectively. Twenty-five patients (15.3%) underwent HPV vaccination after primary treatment. Among the 2833 patients not included in the analysis, only 67 (2.36%) underwent adjuvant vaccination for HPV. Indeed, no impact on recurrence has been detected in this group as well.

Positive ectocervical and endocervical margins were observed in 119 (73%) and 44 (27%) women, respectively. No patient had both margins positive. Table 1 displays the main characteristics of the study population. Overall, 17 (10.4%) patients developed a CIN2+ recurrence during the 5-year follow-up.

### Risk Factors for Recurrence

We evaluated risk factors for developing recurrence after conization over the 5-year follow-up period. Figure 2 shows the risk of developing persistent/recurrent CIN2+ across various populations.

Appendix A shows the main characteristics of patients with CIN2 (*n* = 61) and CIN3 (*n* = 102). The prevalence of endocervical margins was higher in the CIN3 instead of the CIN2 group. Appendix A shows the main characteristics of patients with positive ectocervical and endocervical margins. Patients with endocervical margins were more likely to become infected by HR-HPV types in comparison to patients with ectocervical margins. The main characteristics of the study population related to HPV infection type(s) are reported in Appendix A.

We evaluated the risk of recurrence via univariate and multivariate models. Via univariate analysis, diagnosis of CIN3 instead of CIN2 (HR: 4.88 (95%CI: 1.10, 12.41); *p* = 0.035) and positive endocervical instead of ectocervical margins (HR: 6.44 (95%CI: 2.80, 9.65); *p* < 0.001) are associated with an increased risk of persistence/recurrence (Table 2).

Via multivariate analysis, only positive endocervical instead of ectocervical margins (HR: 4.56 (95%CI: 1.23, 7.95); ***p*** = 0.021) were associated with worse outcomes (Table 3).

## 4. Discussion

The present paper evaluated the outcomes of patients at high risk of recurrence following primary treatment of cervical intraepithelial neoplasia. High-risk patients are considered patients with positive cervical margins and HR-HPV infection persistence. This study is the first to our knowledge to assess the contextual presence of positive margins after surgical treatment and HPV persistence. It reports interesting findings. First, among women treated with primary conization, the high-risk population (those with both positive cervical margins and HR-HPV persistence) accounts for about 5%. Second, positive margins after treatment, both ecto- and endocervical, were more frequent in patients with CIN 3 than in patients with CIN 2 at final histology. Third, also among women with persistent HR-HPV infection, patients with positive endocervical lesions have a higher recurrence rate than patients with ectocervical lesions. Fourth, the use of HPV vaccination did not improve the outcomes of the high-risk group.

Recurrent cervical dysplasia might cause severe issues to women’s health, and it is associated with a significant risk of cervical cancer, and in women willing to preserve their childbearing potential, the execution of multiple conizations might affect fertility and reproductive outcomes. For this reason, surveillance of these patients is delicate, and HPV testing and co-testing, showing little difference in detecting recurrent CIN2+, could be used. Persistence of HR-HPV infection, especially HPV 16/18, are considered independent predisposing factors for recurrence/progression of CIN2+ and cervical cancer. Accumulating evidence suggests that positive cervical margins and HPV persistence are the main risk factors impacting the risk of recurrence [6,16,17,22,23]. Recently, our study group evaluated the correlation between those factors and the risk of recurrence/persistence of cervical dysplasia in the largest Italian study investigating the impact of different surgical techniques for conization in ~3000 women with cervical dysplasia with a 5-year follow-up, finding that HPV persistence is considered to be the only factor associated with an increased risk of recurrence, regardless type of surgical approach [6]. A recent retrospective study with the largest sample of patients (4369 patients who had conization for CIN 2/3+) confirmed the same result: 22% of treated patients with positive margins had a disease recurrence. Furthermore, women with positive margins had more severe pathology at baseline diagnosis, suggesting the possibility of a wide excision to avoid residual lesions in women affected by CIN3 with respect to CIN2 [19]. Previous studies identified as independent risk factors for positive margins an age >35 years, menopause, HSIL in preoperative assessment and colposcopy lesions involving four quadrants [22]. Recurrent CIN2+ was detected in nearly 10.4% of women in our sample, which was higher than the overall average of approximately 6–7% in recent studies [19]; this is probably due to the high percentage of CIN3 patients in our sample (62.6%), to the long median follow-up time and to two combined risks of recurrence, positive margins and HPV persistence.

A growing body of the literature reported data reflecting positive margins. Complete excision of cervical lesions is the main goal to be achieved in surgical treatment, but full excision rates are about 85–90% [24,25]. Yung-Taek Ouh et al. analyzed data from 398 patients [15]. A total of 154 patients experienced persistence of HPV infection after LEEP or conization in a mean follow-up period of 17.3 months (range 4–48). The risk of persistence was higher for HPV 16 infection with respect to other genotypes; moreover, HPV 16 is the most important genotype also relating to the risk of recurrence [26,27,28,29,30]. A meta-analysis by Arbyn et al. showed that HR-HPV results were more accurate than margin status in predicting recurrence, with higher sensitivity (91% *vs* 56%) and equivalent specificity (84%) [31]. For this reason, HPV-based follow up is recommended, and if the test is positive, a colposcopy is recommended.

A recent retrospective cohort study of 2400 women diagnosed with CIN2+ identified individual risk factors that increase HSIL recurrence rates: endocervical canal length, compromised margins and HIV+; an excised canal length of 1.25 cm or more seems to reduce the recurrence rate [32]. Few studies have analyzed the effect of comorbidities on the disease progression of women with CN2/CIN3, but people with weakened immune function (HIV, immunosuppression) are more susceptible to the infection evolving into high grade lesions and/or cervical cancer [33]. Our study highlighted that BMI could influence the different severity of the lesion (CIN2 vs. CIN3) and the different expression of positive margins (ecto- vs endocervical positive margins).

Some points of the present study that should be considered: (i) Our analysis showed that there is no difference in risk of persistence/recurrence for infection with HPV16/18 and other HR-HPV. Although accumulating data support that those with HPV16/18 are at a high risk of persistence in comparison with other HPV types [28], no studies evaluated this association in women with positive surgical margins. (ii) In our series, the executing of vaccination did not improve patients’ outcomes in the group of women with persistent HR-HPV infection. This latter finding is in line with the previous data reported by our study group [34,35,36]. In a previous investigation looking at “adjuvant” HPV vaccination, we observed that vaccination reduces the risk of recurrence. However, we observed that adjuvant vaccination does not impact the risk of having persistent disease (this is particularly evident in patients with positive margins) [36]. On the other side, results from two clinical trials found that quadrivalent HPV vaccination in women who had previously undergone surgery for HPV-related disease significantly reduced the incidence of subsequent HPV-related disease, including high-grade disease [37]. The accumulation of additional cases or large-scale studies are needed to validate our findings.

Another key concept for reducing the number of precancerous and cancerous lesions associated with HPV infection is the spread of the vaccine among male individuals, especially during adolescence.

Unfortunately, as demonstrated in a recent systematic review and meta-analysis conducted by Amantea et al., in which the data from eight studies on more than 140 thousand men aged between 18 and 30 years have been analyzed, the adherence to vaccination is about 11%, with a percentage of adhesion inversely proportional to the increase in age [38].

In Italy, a group of researchers commented on the recommended vaccine for the prevention of infection with HPV in men, designating the vaccine as the preferred treatment [39]. However, although vaccination with new-generation vaccines (nonavalent or quadrivalent) is understandable; in fact, it offers more wider-spectrum protection than the use of the bivalent vaccine.

However, cross-protective efficacy is controversial, as a 2012 meta-analysis showed that the use of a bivalent vaccine protected against some non-vaccine HPV types [40].

The main weaknesses of the present paper included its retrospective nature and the consequent biases related to its design. Data about specific virus types and possible multiple types of infection were not available. The strengths of our work include the large sample analyzed, the multicenter design and the homogeneity of the patients included. It would be interesting to analyze the comorbidities of these patients in relation to disease progression and long-term obstetrical outcomes. In conclusion, this manuscript analyzes the data of a large sample of patients undergoing conization for cervical dysplasia. In this population, positive endocervical margins after surgical treatment are related to a statistically significant increase in the persistence/recurrence rate. During follow up, the correlation between cervical lesions and HR-HPV genotypes has a fundamental role: high-risk HPV has the highest accuracy in predicting recurrence. Given the higher risk of recurrence with positive margins, wider excisions could be indicated to avoid residual lesions. Timely identification of these high-risk patients enables risk stratification and enables individualized management and follow-up strategies. Further prospective studies are necessary to assess the most appropriate follow-up strategy in women with cervical dysplasia and to better understand the role of vaccination in these risk categories.

## Figures and Tables

**Figure 1 vaccines-11-00698-f001:**
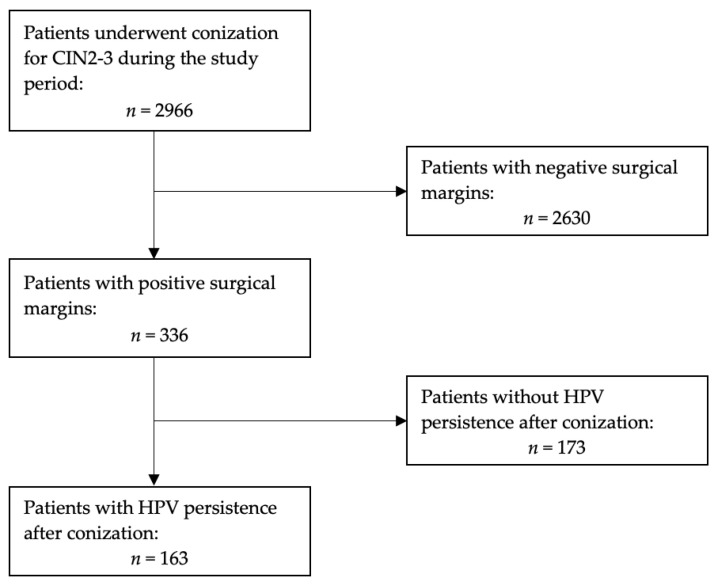
Flow of patients thorough the study design.

**Figure 2 vaccines-11-00698-f002:**
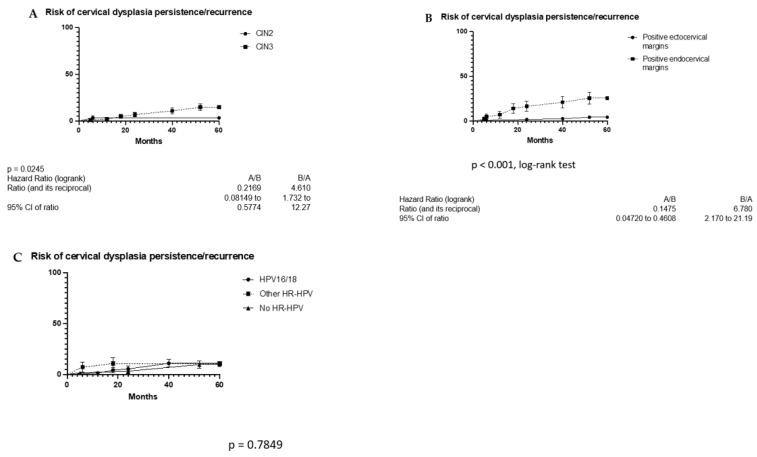
(**A**) Characteristics of patients with CIN2 and CIN3. (**B**) Characteristics of patients with positive ectocervical and endocervical margins. (**C**) Characteristics of the study population related to HPV infection type(s).

**Table 1 vaccines-11-00698-t001:** Characteristics of the study population.

Patients	163
Age (mean)	42.82 (20–74)
BMI (mean)	24.82 (18.4–42)
CIN 2	61 (37.4%)
CIN 3	102 (62.6%)
HPV 16/18	73 (44.7%)
HR-HPV	101 (61.9%)
Laser	2 (1.2%)
LEEP	161 (98.8%)
Ectocervical margin positive	119 (73%)
Endocervical margin positive	44 (27%)
HPV persistence after 6 months	163 (100%)
HPV persistence after 12 months	55 (33.7%)
HPV vaccination after persistence	25 (15.3%)
HPV recurrence	17 (10.4%)

**Table 2 vaccines-11-00698-t002:** Univariate recurrence of cervical dysplasia (CIN2+).

	HR	CI (95%)	*p* Value
Severity of the lesion			
CIN 2	Reference	-	-
CIN 3	4.88	1.10–12.41	0.035
**HPV involved**			
No HR-HPV	Reference	-	-
HPV 16/18	1.58	0.76–4.33	0.928
HR-HPV	1.44	0.89–4.52	0.845
**Surgical approach**			
LEEP	Reference	-	-
LASER	1.32	0.98–2.08	0.156
**Positive margins**			
Ectocervical margins	Reference	-	-
Endocervical positive margins	6.44	2.80–9.65	<0.0001
**HPV vaccination**			
No	References	-	-
Yes	0.85	0.65–1.05	0.108

**Table 3 vaccines-11-00698-t003:** Multivariate recurrence of cervical dysplasia (CIN2+).

	HR	CI (95%)	*p* Value
Severity of the lesion			
CIN2	Reference	-	-
CIN 3	2.897	0.595–14.09	0.188
**Positive surgical margins**			
Ectocervical margins	Reference	-	-
Endocervical margins	4.561	1.235–7.954	0.021

## Data Availability

The data presented in this study are available within the article or Appendix A.

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
