# Peer review of "Outcomes of High-Grade Cervical Dysplasia with Positive Margins and HPV Persistence after Cervical Conization"

_vaccines, 2023, doi:10.3390/vaccines11030698_

Round 1
Reviewer 1 Report
Vaccines- 2162953
REVIEWER 1
The article of Giannini and coauthors, untitled, " Outcomes of high-grade cervical dysplasia with positive margins and HPV persistence after conization " reports the risk factors of recurrence of cervical dysplasia after conization. The strength of this study is the large cohort (2966 patients), included in a multicentric study, the homogeneity of studied population and with a large period of 5-year follow-up. The limitation of this study is its retrospective and multicentric aspects. The authors did not report if there was a centre effect for recurrent dysplasia, CIN3 treated patients, with positive margins or HPV persistence? Also the mid-points during the 5-year follow-up are not described in this study, only a HPV testing at 6-month post-conization is reported in the abstract to evaluate HPV persistence, this 6-month point is not mentioned in the method nor in the result sections. HPV persistence may thus be observed after 6 months but also later after conization, thus were late HPV positive tests also considered is this study? Several HPV methods are also reported in this study, methods may differ depending centres, therefore the authors might observe also method effect? Indeed, detection of HPV DNA vary depending HPV tests, PCR based-methods such as Cobas reported in this study showed lower level of detection compared to non-PCR based methods such as Hybrid capture. In post-treatment follow-up, it is important to use the highest sensitive method, recurrent HPV infection may therefore be more frequently observed in centres using the Cobas method. Another question concerns the HPV genotypes; HPV16/18 are reported in this study with no impact in recurrence (table 2), but the author could not distinguish persistent infection and reinfection post-conization (with other genotypes), persistent HPV infection should involve the same genotype. Were the genotypes identified before conization ? At last, the impact of vaccination combine to conization in dysplasia recurrence and HPV persistence is not clear; it is not related in this study (table 2) but the author in a previous study observed that vaccination reduces the risk of occurrence (lanes 288-289 in discussion section), why such discordant observations ? In this study, impact of vaccination is reported for patients with positive margins and HPV persistence (163 patients among the 2966 included patients), what about the rest of the cohort, were they more vaccinated and thus less at risk of recurrence?
Other remarks:
1. In abstract section, lane 95, HPV persistence is reported at 6 months post-conization and in the method section as the first follow-up visit?
2. In abstract section, lane 99, “overall” is related to the entire cohort of 2966 patients or to the 163 with positive margins and HPV persistence. This term is confusing!
3. In method section, lane 162, time points during the 5-year follow-up are not reported, were they different regarding centres or patients?
4. In method section, lane 164, different methods are reported but there is no information concerning their sensitivity in post-conization HPV testing (in reference 21?). To my knowledge, HC2 does not give HPV16/18 specific results, reported results for HPV16/18 in the table 1 concerns the other methods? If so, no patients with HC2 testing are reported in the study population with positive margins and HPV persistence.
5. In method section, HPV persistence is not defined as in the abstract.
6. In results section, lane 190, 25 patients were vaccinated, I suppose among the 163 patients ? What about the rest of the 2966 cohort, were they more or less vaccinated (impact on dysplasia recurrence?)
7. Figure 1: HPV persistence is written “peristence” twice in the chart. It is not clear from the methods section if Patients without HPV persistence after conization (n=173) were patients with a negative HPV test at 6 months or at some other points during the 5-year follow-up ? Were these patients found negative mostly with HC2 ? What is the risk of CIN2+ recurrence in this sub-group compared to patients with HPV persistence?
8. In results section, lane 196, “overall” concerns CIN2+ recurrence in only the 163 patients, what about the cohort of 2966 (we expected the overall CIN2+ recurrence to be less than 10.4%) ?
9. Table 1 is truncated on 2 pages.
10. Figure 2 contains 3 figures too small to be readable. Figure notes below figures A and B (also unreadable) should be in legend notes with Figure 2 title (there is no title to this figure?). Figure letters A, B, C in legend should be on the same size than at figure levels.
11. Tables 2 and 3 titles should precise Recurrence of cervical dysplasia (CIN2+).
12. In discussion section, lane 231, “accounts for about 5” but 5 what, is that % or 5-fold at risk or else ?
13. In discussion section, lane 257, the higher % of recurrent CIN2+ in this study as claimed by the authors, is also related to that it combined two risks, positive margins and HPV persistence, whereas the other studies regard only one of these risks.
14. In discussion section, lane 267, the reported meta-analysis of M. Arbyn found HPV persistence more accurate than margin status, whereas this study shows reverse statement, only margin status appears in multivariate analysis of recurrence, how the authors discussed this major discordant observation? Their analysis does not include patients with positive margin but negative HPV persistence, thus creating a bias of analysis.
15. In discussion section, lane 285, “vaccination did not improve patients outcome” and in lane 288, “in a previous investigation… vaccination reduces the risk of having persistent disease” and in lane 289 “vaccination does not impact the risk”, what is the good conclusion? Were patients vaccinated also with the quadrivalent vaccine or the nonavalent one? There might be some difference regarding the vaccine used.
Author Response
"Response to Reviewers" for vaccines-2162953
Dear Editor,
Thank you for considering our manuscript entitled “Outcomes of high-grade cervical dysplasia with positive margins and HPV persistence after conization” for publication in “Vaccines”. We have really appreciated the suggestions and corrections made by the reviewer and we have modified the paper accordingly. We are convinced that with the modifications recommended the paper has been significantly improved and we strongly hope it will be considered for publication in your journal. Please find enclosed a revised version of the manuscript, modified on the basis of the reviewer's criticisms.
Reviewer's criticisms and comment were reported in Bold.
REVIEWER 1
The article of Giannini and coauthors, untitled, " Outcomes of high-grade cervical dysplasia with positive margins and HPV persistence after conization " reports the risk factors of recurrence of cervical dysplasia after conization. The strength of this study is the large cohort (2966 patients), included in a multicentric study, the homogeneity of studied population and with a large period of 5-year follow-up. The limitation of this study is its retrospective and multicentric aspects.
Comment 1R1: The authors did not report if there was a centre effect for recurrent dysplasia, CIN3 treated patients, with positive margins or HPV persistence?
Response 1R1: No difference to recurrent dysplasia, CIN3 treated patients, with positive margins or HPV persistence was found between centers. This information is included in the text at lines 174-175.
Comment 2R1: Also, the mid-points during the 5-year follow-up are not described in this study, only a HPV testing at 6-month post-conization is reported in the abstract to evaluate HPV persistence, this 6-month point is not mentioned in the method nor in the result sections. HPV persistence may thus be observed after 6 months but also later after conization, thus were late HPV positive tests also considered is this study?
Response 2R1: All patients included in the study had persistence of the HPV infection demonstrated with control Pap-smear performed 6 months after conization. This point was included in the methods at lines 171-172.
Comment 3R1: Several HPV methods are also reported in this study, methods may differ depending centres, therefore the authors might observe also method effect?
Response 3R1: No significant difference in the positivity rate was observed between the test used to detect HPV infection in each center. This point was included in the methods at lines 174-175.
Comment 4R1: Indeed, detection of HPV DNA vary depending HPV tests, PCR based-methods such as Cobas reported in this study showed lower level of detection compared to non-PCR based methods such as Hybrid capture. In post-treatment follow-up, it is important to use the highest sensitive method, recurrent HPV infection may therefore be more frequently observed in centres using the Cobas method.
Response 4R1: Thanks for the comment. However, due to the multicentric and the retrospective nature of this study, it is not possible to determine which test was used for post-treatment follow-up for each patient. This advice will be used for future studies.
Comment 5R1: Another question concerns the HPV genotypes; HPV16/18 are reported in this study with no impact in recurrence (table 2), but the author could not distinguish persistent infection and reinfection post-conization (with other genotypes), persistent HPV infection should involve the same genotype. Were the genotypes identified before conization?
Response 5R1: All patients were screened to identify HPV genotype(s) before conization. This point was included in the methods at lines 194-195.
Comment 6R1: At last, the impact of vaccination combine to conization in dysplasia recurrence and HPV persistence is not clear; it is not related in this study (table 2) but the author in a previous study observed that vaccination reduces the risk of occurrence (lanes 288-289 in discussion section), why such discordant observations?
Response 6R1: As reported in the text in lines 303-305, the results obtained by our working group agree in demonstrating that adjuvant vaccination has no effect on the persistence of HPV infection.
Comment 7R1: In this study, impact of vaccination is reported for patients with positive margins and HPV persistence (163 patients among the 2966 included patients), what about the rest of the cohort, were they more vaccinated and thus less at risk of recurrence?
Response 7R1: Of the 2833 patients not included in the analysis, only 67 carried out adjuvant vaccination for HPV. No impact on recurrence has been detected in this group as well. This point was included in the methods at lines 202-204.
Comment 8R1: In abstract section, lane 95, HPV persistence is reported at 6 months post-conization and in the method section as the first follow-up visit?
Response 8R1: All patients included in the study had persistence of the HPV infection demonstrated with control Pap-smear performed 6 months after conization. This point was included in the methods at lines 150-151.
Comment 9R1: In abstract section, lane 99, “overall” is related to the entire cohort of 2966 patients or to the 163 with positive margins and HPV persistence. This term is confusing!
Response 9R1: The term "overall" has been changed to "Of 163 patients included". This point was included in the abstract at line 99.
Comment 10R1: In method section, lane 162, time points during the 5-year follow-up are not reported, were they different regarding centres or patients?
Response 10R1: No significant difference between the centers was found. All patients underwent an outpatient follow-up at 6 and 12 months after surgery. This point was included in the methods at lines 171-172.
Comment 11R1: In method section, lane 164, different methods are reported but there is no information concerning their sensitivity in post-conization HPV testing (in reference 21?). To my knowledge, HC2 does not give HPV16/18 specific results, reported results for HPV16/18 in the table 1 concerns the other methods? If so, no patients with HC2 testing are reported in the study population with positive margins and HPV persistence.
Response 11R1: “Qiagen HC2 test is a sandwich capture molecular hybridization assay: it is a signal amplification detection method based on chemiluminescence that detects 13 HR HPV types: HPV 16, 18, 31, 33, 35, 39, 45, 51, 52, 56, 58, 59, and 68” from reference 21 (PMID: 36292048). Due to the retrospective nature of this study, it is not possible to determine which test was used for post-treatment follow-up for each patient.
Comment 12R1: In method section, HPV persistence is not defined as in the abstract.
Response 12R1: No definition of persistence was reported in the abstract. In method section, instead, persistence was defined as the detection of high-grade cervical lesion at the time of first follow-up visit (lines 177-178)
Comment 13R1: In results section, lane 190, 25 patients were vaccinated, I suppose among the 163 patients? What about the rest of the 2966 cohort, were they more or less vaccinated (impact on dysplasia recurrence?)
Response 13R1: Of the 2833 patients not included in the analysis, only 67 carried out adjuvant vaccination for HPV. No impact on recurrence has been detected in this group as well. This point was included in the methods at lines 202-204.
Comment 14R1: Figure 1: HPV persistence is written “peristence” twice in the chart. It is not clear from the methods section if Patients without HPV persistence after conization (n=173) were patients with a negative HPV test at 6 months or at some other points during the 5-year follow-up? Were these patients found negative mostly with HC2? What is the risk of CIN2+ recurrence in this sub-group compared to patients with HPV persistence?
Response 14R1: All patients included in the study had persistence of the HPV infection demonstrated with control Pap-smear performed 6 months after conization. This point was included in the methods at lines 150-151. Typo error “peristence" has been corrected in the figure 1.
Comment 15R1: In results section, lane 196, “overall” concerns CIN2+ recurrence in only the 163 patients, what about the cohort of 2966 (we expected the overall CIN2+ recurrence to be less than 10.4%) ?
Response 15R1: Of the 2833 patients not included in the analysis, only 67 carried out adjuvant vaccination for HPV. No impact on recurrence has been detected in this group as well. This point was included in the methods at lines 202-204.
Comment 16R1: Table 1 is truncated on 2 pages.
Response 16R1: Unfortunately, the table is truncated on two pages due to the layout of the journal.
Comment 17R1: Figure 2 contains 3 figures too small to be readable. Figure notes below figures A and B (also unreadable) should be in legend notes with Figure 2 title (there is no title to this figure?). Figure letters A, B, C in legend should be on the same size than at figure levels.
Response 17R1: Figure 2 has been modified to make the figures and the legends more readable. Figure letters A, B, C in the legend have been modified.
Comment 18R1: Tables 2 and 3 titles should precise Recurrence of cervical dysplasia (CIN2+).
Response 18R1: Tables’ 2 and 3 titles have been modified.
Comment 19R1: In discussion section, lane 231, “accounts for about 5” but 5 what, is that % or 5-fold at risk or else?
Response 19R1: Lines 244-246 were modified: “Among women treated with primary conization, the high-risk population (those with both positive cervical margins and HR-HPV persistence) accounts for about 5%”.
Comment 20R1: In discussion section, lane 257, the higher % of recurrent CIN2+ in this study as claimed by the authors, is also related to that it combined two risks, positive margins and HPV persistence, whereas the other studies regard only one of these risks.
Response 20R1: Thank you for your suggestion. This consideration has been added in the text in lines 282-283.
Comment 21R1: In discussion section, lane 267, the reported meta-analysis of M. Arbyn found HPV persistence more accurate than margin status, whereas this study shows reverse statement, only margin status appears in multivariate analysis of recurrence, how the authors discussed this major discordant observation? Their analysis does not include patients with positive margin but negative HPV persistence, thus creating a bias of analysis.
Response 21R1: The meta-analysis analyzed in our study reports that the presence of a high-risk HPV post-treatment predicts the residual or recurrent CIN2+ with higher accuracy than the status of the resection margins.
Comment 22R1: In discussion section, lane 285, “vaccination did not improve patients outcome” and in lane 288, “in a previous investigation… vaccination reduces the risk of having persistent disease” and in lane 289 “vaccination does not impact the risk”, what is the good conclusion? Were patients vaccinated also with the quadrivalent vaccine or the nonavalent one? There might be some difference regarding the vaccine used.
Response 22R1: Lines: 310-315: In our series the executing of vaccination did not improve patients’ outcomes in the group of women with persistent high risk-HPV infection. This latter finding is in line with previous data reported by our study group. In a previous investigation looking at “adjuvant” HPV vaccination, we observed that vaccination reduces the risk of recurrence. However, we observed that adjuvant vaccination does not impact the risk of having persistent disease.
About the vaccine used, due to the retrospective nature of this study, it is not possible to determine which vaccine was used for adiuvant vaccination for each patient. This advice should be used for future studies.
We thank the reviewer for the accurate revision of the paper and for the precise comments. The manuscript has been edited as suggested.
Yours sincerely,
Thanks again for your availability and consideration.
Prof. Giorgio Bogani, M.D., Ph.D,
University La Sapienza of Rome
Fondazione IRCCS Istituto Nazionale dei Tumori di Milano Via Venezian 1, 20133 Milan, Italy
Phone: 00390223902392 Fax 00390223902349.
Reviewer 2 Report
This is a very interesting clinical following up study. In this study, it included a lot samples, and lasting for a long time. I am sure that the readers will be sparked from this study.
However, I still have some suggestions to help the readers to understand CIN and HPV better. Because, it is a clinical associated study, while many scientists do not have good pathological experience.
1. Could you please add representative Histological Pictures.
2. Could you please provide some visulized data to show the differences of "Before" and "After" conization, even Recurrence included, if possible.
3. The resolution of Fig.2 have to be upgraded, what I read is very weak.
Thank you!
Author Response
"Response to Reviewers" for vaccines-2162953
Dear Editor,
Thank you for considering our manuscript entitled “Outcomes of high-grade cervical dysplasia with positive margins and HPV persistence after conization” for publication in “Vaccines”. We have really appreciated the suggestions and corrections made by the reviewer and we have modified the paper accordingly. We are convinced that with the modifications recommended the paper has been significantly improved and we strongly hope it will be considered for publication in your journal. Please find enclosed a revised version of the manuscript, modified on the basis of the reviewer's criticisms.
Reviewer's criticisms and comment were reported in Bold.
REVIEWER 2
This is a very interesting clinical following up study. In this study, it included a lot samples, and lasting for a long time. I am sure that the readers will be sparked from this study.
However, I still have some suggestions to help the readers to understand CIN and HPV better. Because, it is a clinical associated study, while many scientists do not have good pathological experience.
Comment 1R2: Could you please add representative Histological Pictures.
Response 1R2: Unfortunately, in our opinion the addition of histological pictures would not increase the quality of our work. Thanks for the advice.
Comment 2R2: Could you please provide some visulized data to show the differences of "Before" and "After" conization, even Recurrence included, if possible.
Response 2R2: In our opinion the addition of visulized data would not increase the quality of our work. Thanks for the advice.
Comment 3R2: The resolution of Fig.2 have to be upgraded, what I read is very weak.
Response 3R2: Figure 2 has been modified to make the figures and the legends more readable. Figure letters A, B, C in the legend have been modified.
We thank the reviewer for the accurate revision of the paper and for the precise comments. The manuscript has been edited as suggested.
Yours sincerely,
Thanks again for your availability and consideration.
Prof. Giorgio Bogani, M.D., Ph.D,
University La Sapienza of Rome
Fondazione IRCCS Istituto Nazionale dei Tumori di Milano Via Venezian 1, 20133 Milan, Italy
Phone: 00390223902392 Fax 00390223902349.